# *N*-glycosylation in the Pre-Membrane Protein Is Essential for the Zika Virus Life Cycle

**DOI:** 10.3390/v12090925

**Published:** 2020-08-23

**Authors:** Yong-Dae Gwon, Eva Zusinaite, Andres Merits, Anna K. Överby, Magnus Evander

**Affiliations:** 1Department of Clinical Microbiology, Virology, Umeå University, 901 85 Umeå, Sweden; kwon.yongdae@umu.se (Y.-D.G.); anna.overby@umu.se (A.K.Ö.); 2Umeå Centre for Microbial Research (UCMR), Umeå University, 901 85 Umeå, Sweden; 3Institute of Technology, University of Tartu, 50411 Tartu, Estonia; eva.zusinaite@gmail.com (E.Z.); andres.merits@ut.ee (A.M.); 4The Laboratory for Molecular Infection Medicine Sweden (MIMS), Umeå University, 901 85 Umeå, Sweden

**Keywords:** Zika virus, *N*-glycosylation, pre-membrane, envelope, virus life cycle

## Abstract

Asparagine (N)-linked protein glycosylation plays an important role in protein synthesis and modification. Two Zika virus (ZIKV) structural proteins, the pre-membrane (prM) and envelope (E) protein are *N*-glycosylated. The prM protein of all ZIKV strains contains a single *N*-linked glycosylation site, while not all strains contain an N-linked site in the E protein. Our aim was to examine the impact of prM and E N-linked glycosylation on ZIKV infectivity and cell trafficking. Using a ZIKV infectious clone, we found that when the *N*-glycan sites were removed, the prM- and the prM/E-double mutants did not produce an infectious virus in the supernatant. Further, by using ZIKV prME constructs, we found that *N*-glycosylation was necessary for effective secretion of ZIKV virions. The absence of the *N*-glycan on prM or E caused protein aggregation in the rough endoplasmatic reticulum (ER) compartment. The aggregation was more pronounced for the prM-mutation, and the mutant virus lost the ER-Golgi intermediate compartment (ERGIC) localization. In addition, lack of the *N*-glycan on prM induced nuclear translocation of CCAAT-enhancer-binding protein homologous protein (CHOP), an ER stress marker. To conclude, we show that the prM *N*-glycan is essential for the ZIKV infectious cycle, and plays an important role in viral protein trafficking, protein folding, and virion assembly**.**

## 1. Introduction

Annually, millions of people suffer from the emergence or re-emergence of flaviviruses [1]. Zika virus (ZIKV) is an emerging flavivirus (from the family of *Flaviviridae*) transmitted primarily by *Aedes* mosquitoes [2]. In most cases, ZIKV causes no symptoms or a dengue-like illness, with rashes, conjunctivitis, and other mild clinical manifestations [3,4]. However, the outbreak of ZIKV in French Polynesia and the Americas showed that the virus was associated with clinical manifestations such as microcephaly and Guillain-Barré syndrome [5].

ZIKV has a ~10.8 kb positive-sense single-stranded RNA genome which encodes a single polyprotein that after processing produces ten mature proteins; three structural (capsid, C; pre-membrane, prM; and envelope, E), and seven non-structural (NS1, NS2A, NS2B, NS3, NS4A, NS4B, and NS5) proteins [6]. The structural proteins play important roles in the virus life cycle, particularly attachment, entry into host cells, virion formation, maturation, and release of the matured virion [7]. The E protein is involved in binding to the host cell surface [8]. In immature viral particles, the prM protein forms a heterodimer with the E protein [9,10]. For virus maturation, cleavage of the capsid protein, in a trans-Golgi compartment, triggers prM cleavage to a mature membrane protein M and pr by furin, resulting in mature and infectious particles [10,11,12,13].

N-linked glycosylation is one of the most common types of membrane protein modification and involves the transfer of oligosaccharides to the asparagine (N) residue present in the conserved glycosylation motif N-X-S/T and takes place in the endoplasmic reticulum (ER), followed by further modifications as the protein travels through the secretory pathway [14]. Many viral envelope proteins contain N-linked glycans, which can play major roles in correct folding, physicochemical properties, intracellular transport, viral entry, and immune evasion [15,16]. 

Phylogenetically, there are two major lineages of ZIKV isolates, the African and Asian lineages, where the African lineage is the ancestral root to the Asian lineage [17,18]. The genetic difference between the lineages is up to 4% in their amino acid sequences [19,20]. Normally, ZIKV contains N-linked glycosylation (*N*-glycan) motifs in their structural proteins [21]. For the E protein, the *N*-glycan motif is absent in many of the African ZIKV isolates, but present in ZIKV strains (Asian lineages) from recent outbreaks [22]. Recently accumulated evidence revealed that the E protein *N*-glycan in ZIKV plays important roles in virion assembly and infectivity [23,24], viral tissue tropism [25], and pathogenesis [6,17]. 

The prM protein *N*-glycan in flaviviruses seems to be important for multiple stages of flavivirus biology. Several reports suggest a multifaceted role for the prM *N*-glycan in the Japanese encephalitis virus (JEV) [26], West Nile fever virus (WNV) [27], dengue virus (DENV) [28], and tick-borne encephalitis virus (TBEV) [29]. On the other hand, the role of the prM protein *N*-glycan in ZIKV is still unclear.

In the present study, we used a reverse genetic system with a genetically stable reporter, based on a Brazilian ZIKV isolate, to address the role of the N-linked glycosylation motifs within the prM (69NTTS73) and the E protein (154NDTG157). We found that ZIKV with a single amino acid substitution (N69Q) in the *N*-glycosylation site of prM, or a double amino acid substitution (N69Q and N154Q) in both prM and E, were not infectious in mammalian cells. Furthermore, we found that a prME construct with an *N*-glycosylation site mutation in prM did not promote secretion of virus-like particles and induced expression of the CCAAT-enhancer-binding protein homologous protein (CHOP) transcription factor. Our results increased knowledge regarding the role of the N-linked glycosylation of the prM protein in the ZIKV life cycle.

## 2. Materials and Methods

### 2.1. Cells

Vero B4 cells (African green monkey kidney) [30] were maintained at 37 °C, 5% CO_2_ in Dulbecco’s modified Eagle medium (DMEM; Thermo Scientific, Waltham, MA, USA) supplemented with 5% fetal bovine serum (FBS; Thermo Scientific, Waltham, MA, USA), 2% 4-(2-hydroxyethyl)-1-piperazineethanesulfonic acid (HEPES; Thermo Scientific, Waltham, MA, USA), 10 units/mL penicillin and 10 µg/mL streptomycin (Thermo Scientific, Waltham, MA, USA). For virus infection, cell maintenance medium was used containing the same components, except at a lower FBS concentration (2%).

### 2.2. Mutagenesis of the N-glycan Motif, Plasmid Constructions, and Rescue of Infectious Clones

To introduce the mutations in the *N*-glycan motif in the ZIKV structural proteins, an overlapping polymerase chain reaction (PCR) strategy was designed as illustrated in Figure 1a. From the infectious clone of the full-length Brazilian ZIKV isolate (icZIKV, BeH819015, Genbank: KU365778, 2015), the genes of the structural proteins were cloned into a pJET1.2 vector (Thermo Scientific, Waltham, MA, USA). Three sets of primers were used to generate mutations in the *N*-glycan motifs of prM and E, either in one of the motifs or both (termed N69Q, N154Q, or N69Q/N154Q hereafter). Amplified PCR fragments were stitched and ligated into the pJET1.2 vector and sequences of obtained constructs were verified by Sanger-sequencing. The variants of prM and E genes were sub-cloned into an infectious clone of the Brazilian ZIKV that contained gene encoding for the *Zoanthus sp.* green fluorescence protein (ZsGreen), hereafter termed the wild-type (WT) clone, using a XhoI-AvrII restriction enzymes (Thermo Scientific, Waltham, MA, USA); sequences were confirmed by Sanger-sequencing (Figure 1b, Table 1).

Rescue of the virus from WT clone and its mutant variants was carried out using a similar approach to that previously described [31]. Briefly, five micrograms of the four ZIKV constructs were linearized using AgeI restriction enzyme. The resulting DNA fragments were purified using NucleoSpin Gel and PCR Clean-up kit (Macherey-Nagel, Dueren, Germany) and eluted with 6 µL water. The mMESSAGE mMACHINE SP6 transcription kit (Thermo Scientific, Waltham, MA, USA) was used to synthesize in vitro transcribed 5′ capped RNA from 3 µL of cDNA in a 20 µL reaction volume. The transcripts were used to transfect 2 × 10^5^ of pre-seeded Vero B4 cells using TransIT^®^-mRNA Transfection Kit (Mirus Bio, Madison, WI, USA). Four hours post-transfection (hpt), the cells were washed five times with phosphate-buffered saline (PBS) to minimize the contamination in RT-PCR from residues of in vitro transcripted ZIKV RNA, and then fresh DMEM was added. Transfected cells were monitored for fluorescence of the expressed ZsGreen reporter and supernatants were harvested for RNA isolation.

### 2.3. Immunofluorescence Microscopy

For indirect immunofluorescence microscopy, Vero B4 cells were grown on eight-well chamber slides (Sarstedt, Nümbrecht, Germany). At 50% confluence, cells were transfected with in vitro transcribed ZIKV RNA or pcDNA3.1-prME plasmids and incubated for 72 h (ZIKV RNA) or 48 h (pcDNA3.1-prME plasmid). The cells were washed with PBS, fixed with 4% paraformaldehyde, and permeabilized with 0.02% Triton- X-100 for 5 min. 

To detect ZIKV E-protein expression, samples were stained with rabbit anti-ZIKV E protein antibody (ZEND20-A; Alpha Diagnostic, San Antonio, TX, USA) diluted in 5% FBS/Dulbecco PBS. To detect the CHOP marker, samples were stained with mouse anti-CHOP antibody (Thermo Scientific, Waltham, MA, USA). To detect the rough ER, samples were visualized by using a mouse anti-Sec61 alpha antibody (Abcam, Cambridge, UK). The ER-Golgi intermediate compartment (ERGIC) was detected by using the mouse anti-ERGIC53 antibody (Santa Cruz Biotechnology, Dallas, TX, USA). Incubation with primary antibodies was followed by three washes and incubation with secondary anti-rabbit antibody conjugated to Alexa Fluor 488 or an anti-mouse conjugated to Alexa Fluor 568 (Thermo Scientific, Waltham, MA, USA). Nuclei were counterstained 300 nM 4′,6-diamidino-2-phenylindole (DAPI) in PBS.

For the CHOP expression study, cells were treated with brefeldin A (1 µg/mL; Thermo Scientific, Waltham, MA, USA) or tunicamycin (1 µg/mL; Thermo Scientific, Waltham, MA, USA) for 24 h before transfection, as a control for CHOP expression. In addition, the ZsGreen reporter signal was detected from the reporter protein embedded in the ZIKV genome. The samples were analyzed using a Nikon A1R+ (Nikon, Tokyo, Japan) or a Zeiss 710 (Carl Zeiss, Oberkochen, Germany) confocal microscope and ImageJ 1.52p software.

### 2.4. Plaque Forming Assay and Virus Titration

In order to determine the infectivity of rescued viruses, a plaque-forming assay was performed. Briefly, Vero B4 cells (2 × 10^5^ cells per well) were seeded in 12-well plates one day prior to infection. Cells were infected with 100 µL of rescued viruses at different dilutions for 1 h at 37 °C and washed. Thereafter, 2% carboxymethyl cellulose (CMC; Sigma Aldrich, St. Louis, MO, USA) in DMEM was added to cells for 72 h at 37 °C with 5% CO2. The plates were fixed with 4% paraformaldehyde in neutral buffer and stained with 1% crystal violet solution (1% crystal violet powder, 20% *w*/*v* methanol in distilled H_2_O).

In parallel, the same above protocol was used for virus titration except viruses were 10-fold serial diluted (from 10^−1^ to 10^−5^). Visible plaques were counted and the viral titers (plaque-forming unit (pfu)/mL) were calculated.

### 2.5. Real-Time RT-PCR for the Detection of ZIKV RNA

Growth kinetics of the different ZIKV constructs were analyzed by real-time RT-PCR. Viral RNA was isolated from 100 µL of supernatant by using a viral RNA isolation kit (Macherey-Nagel, Dueren, Germany). A first-strand synthesis kit, with ZIKV specific primers, was used to synthesize the cDNA from isolated RNA (Thermo Scientific, Waltham, MA, USA). To perform real-time qPCR, synthesized cDNA was used as a template in a mixture of SYBR master mix (Sigma Aldrich, St. Louis, MO, USA) and ZIKV NS5 specific primers (Forward: GTACATGGACTACCTATCCACC, Reverse: CTGACTAGCAGGCCTGACAAC). The qPCR reaction was carried out using the StepOnePlus™ Real-Time PCR system (Thermo Scientific, Waltham, MA, USA). Obtained cycle threshold (Ct) values were converted to copy number of ZIKV NS5 RNA by using a standard curve. This was generated by cloning the 105 bp long ZIKV NS5 PCR amplicon into a pJET1.2 vector (Thermo Scientific, Waltham, MA, USA), and prepared dilutions from 10^9^ to 10^1^. Ct values were transformed to a standard curve by using GraphPad Prism 7.0 software (GraphPad Software, San Diego, CA, USA).

### 2.6. pcDNA3.1-prME Plasmid Construction

To construct an expression plasmid for ZIKV prME, the prME gene from the plasmids used in Section 2.2. (capsid-prME genes in the pJET1.2 vector) were PCR amplified using the primer set Forward:5′-CAGCTAGCATGGGCGCCGATACCTCCGTGGGCATT-3′, Reverse:5′- TAGTTTAAACTTAAGCAGAGACGGCTGTGGA-3′. The PCR fragments were inserted into the pcDNA3.1 vector (Thermo Scientific, Waltham, MA, USA) under a human cytomegalovirus (CMV) promoter using NheI-PmeI digestion/T4 DNA ligation (Thermo Scientific, Waltham, MA, USA). Sequences of all for pcDNA3.1-prME constructs were confirmed by Sanger-sequencing.

### 2.7. Glycosidase Treatment

In order to confirm the absence or presence of an *N*-glycan in the prM and E proteins, peptide N-glycosidase F (PNGase F; NEB, Ipswich, MA, USA) treatment was carried out using Vero B4 cell lysates that were either: (1) infected by different ZIKV constructs or (2) transfected by different ZIKV prME expression plasmids. 

For ZIKV infection, Vero B4 cells (2 × 10^5^ cells) were seeded in twelve-well plates one day prior to infection. Cells were infected with 100 µL of rescued viruses for 1 h at 37 °C and washed. After 72 h, cell lysis and protein extraction were performed. The protein samples were mixed with 1× glycoprotein denaturing buffer (0.5% sodium dodecyl sulfate, 40 mM dithiothreitol) and heated at 95 °C for 10 min. After a short spin, the PNGase F reaction was carried out in the presence of 1× glycosidase reaction buffer (50 mM sodium phosphate, pH 7.5) and 1% NP-40 for 1 h at 37 °C. Subsequently, the samples were mixed with 4× NuPAGE LDS Sample Buffer (Thermo Scientific, Waltham, MA, USA) and heated at 95 °C for 5 min, and proteins were resolved by SDS-PAGE followed by Western blotting. For the Western blotting, proteins transferred to PVDF (Polyvinylidene difluoride) membranes (GE Healthcare, Chicago, IL, USA) were stained with rabbit anti-ZIKV E protein antibody, rabbit anti-ZIKV prM protein antibody (ZPRM11-S; Alpha Diagnostic, San Antonio, TX, USA) or mouse anti-β actin antibody (Santa Cruz Biotechnology, Dallas, TX, USA), diluted in 1x PBST (1X phosphate-buffered saline with 0.1 % Tween^®^ detergent) with 5% nonfat dry milk. Incubation with primary antibodies was followed by three washes and incubation with secondary anti-rabbit or anti-mouse conjugated to HRP (Horseradish peroxidase, Thermo Scientific, Waltham, MA, USA).

For ZIKV prME plasmid transfection, Vero B4 cells (5 × 10^5^ cells) were seeded in six-well plates one day prior to transfection. Cells were transfected with 2.5 µg of plasmids by using the Lipofectamine 3000 reagent (Thermo Scientific, Waltham, MA, USA) following the manufacturer’s protocol. After 48 h, cell lysis and protein extraction were performed. PNGase F treatment and western blotting were carried out as above.

### 2.8. Measurement of Secreted E Protein

Vero B4 cells (2 × 10^6^) were seeded in 25T flasks and 24 h later 5 µg of pcDNA3.1-prME plasmids were transfected to the cells using the Lipofectamine 3000 reagent (Thermo Scientific, Waltham, MA, USA) following the manufacturer’s protocol. The supernatants and cells were harvested for Western blotting 48 h post-transfection. To collect secreted prME proteins in the supernatant, we performed ultracentrifugation at 100,000× *g* for 1.5 h and resuspended the pellet in 2× Laemmli buffer (4% SDS, 20% glycerol, 120 mM Tris-HCl pH 6.8, 0.02% bromophenol blue). Proteins were separated using SDS-PAGE and blotted onto PVDF membranes. Proteins on the membranes were detected using rabbit anti-ZIKV E protein antibody or mouse anti-β actin (as an internal control for data analysis) antibody (Santa Cruz Biotechnology, Dallas, TX, USA) diluted in 1x PBST with 5% nonfat dry milk. Incubation with primary antibodies was followed by three washes and incubation with secondary anti-rabbit or anti-mouse conjugated to HRP (Thermo Scientific, Waltham, MA, USA). Quantification of the band intensity was performed using ImageJ software.

### 2.9. Comparison of the N-glycan Motif in the Flavivirus Genus

To collect information regarding the *N*-glycan motif in different flaviviruses, we searched one or two representative strains of the flavivirus genus from the PubMed or the UniProt database [32].

The list of analyzed flaviviruses contains WNV (strain ArB3573/82, GenBank accession: DQ318020), WNV (strain NY-99, Entry code: Q9Q6P4), JEV (strain SA-14, Entry code: P27395), Saint Louis encephalitis virus (strain MS1-7, Entry code: P09732), DENV type 1 (strain Nauru/West Pac/1974, Entry code: P17763), DENV type 2 (strain Thailand/16681/1984, Entry code: P29990), DENV type 3 (strain Sri Lanka/1266/2000, Entry code: Q6YMS4), DENV type 4 (strain Singapore/8976/1995, Entry code: Q5UCB8), ZIKV (strain MR766, Entry code: Q32ZE1), ZIKV (isolate ZIKV/Human/French Polynesia/10087PF/2013, Entry code: A0A024B7W1), Yellow fever virus (strain 17D vaccine, Entry code: P03314), Yellow fever virus (isolate Ivory Coast/1999, Entry code: Q6J3P1), TBEV (strain Hypr, Entry code: Q01299), and Powassan virus (strain LB, Entry code: Q04538).

### 2.10. Data Analysis

Means and standard deviations (SD) were calculated with GraphPad Prism 7.0 software. All statistical analyses were performed using two-tailed t-test by GraphPad Prism 7.0 software; *p* < 0.05 was considered statistically significant.

## 3. Results

### 3.1. Removal of the N-glycan Site in ZIKV prM Resulted in Impaired Viral Gene Expression and No Infectious Virus

To explore the role of the *N*-glycosylation in the ZIKV pre-membrane or envelope proteins, we used site-directed mutagenesis to introduce a single/double amino acid substitution (N69Q, N154Q, or N69Q/N154Q) in a previously described ZIKV infectious clone containing a reporter gene (Figure 1a) [31]. The expression of the reporter gene and the ZIKV E protein were analysed 72 h after transfection by using an immunofluorescence assay. All used ZIKV variants showed co-localization of both ZsGreen and ZIKV E protein expression in positive cells, indicating that the reporter gene was a good proxy for viral protein synthesis and the site of viral infection (Figure 2).

Then we addressed how the mutations in the ZIKV *N*-glycan motifs affected the spread from transfected cells to neighbouring cells as monitored by ZsGreen marker expression, following the schedule in Figure 3a. The cultures transfected with RNAs of WT clone and N154Q showed spread of marker expression (Figure 3b). However, we did not observe spread of marker expression for ZIKV harbouring N69Q and N69Q/N154Q mutations (Figure 3b). From this result, we concluded that the lack of prM *N*-glycosylation in ZIKV negatively affected the virus spread from the host cell, which suggested a lack of progeny virus production or/and inability of progeny virions to infect naïve cells.

We then analyzed the kinetics of viral egress, here defined as viral progeny being produced from the cells, by measuring the virus RNA copy numbers in the harvested supernatant using a ZIKV NS5 gene-specific qRT-PCR, following the schedule in Figure 3a. In comparison to WT virus, the N154Q virus RNA reached a plateau on day 4. The RNA copy numbers of N69Q and N69Q/N154Q viruses showed an increase of virus RNA copy numbers at two days post-transfection but decreased thereafter (Figure 3c), most likely indicating absence of infectious virion production. This showed that prM *N*-glycosylation was important for ZIKV release and spread.

Because the N69Q and N69Q/N154Q substitutions blocked the spread of virus in the cell culture and the release of virus RNA, we decided to study the infectivity of harvested progeny viruses from seven days post-transfection by analysing the plaque-forming ability. We added the harvested supernatants to a new batch of Vero B4 cells and analysed the plaque-forming four days post-infection. The plaques from the WT clone infection resembled the plaques from control infections by Brazilian ZIKV and African ZIKV (Figure 3d). Supernatant harvested from cells transfected with a mutant harbouring the N154Q substitution was able to form plaques, but the plaques were smaller than for the WT, and the virus titer was 49 fold lower (Figure 3d). Noticeably, supernatants from cells transfected with the RNAs of constructs harbouring N69Q and N69Q/N154Q substitutions lacked the ability to produce detectable plaques. This suggested that removal of the *N*-glycosylation site of prM in ZIKV muted infectivity, in line with the lack of ZsGreen spread and inhibition of virus RNA egress (Figure 3b–d). 

To verify that the mutation in the *N*-glycan site resulted in absence of the *N*-glycan, we infected Vero B4 cells with the viruses producing viable progeny. Three days post-infection, we harvested the cell lysate and detected prM and E proteins with and without PNGase F treatment. PNGase F is one of the glycosidases that cleave the *N*-glycan from the protein. The molecular mass for the native prM with an *N*-glycan is between 1.4–2.7 kDa [33]. The WT clone-derived virus-infected cells showed a downshift of the molecular mass after PNGase F treatment for both the prM and E protein, indicating that both proteins were N-glycosylated. The N154Q virus-infected cells and the African ZIKV (MR766) showed no difference in molecular mass after PNGase F treatment for the E protein (Figure 3e).

In summary, the consequence of removing the *N*-glycan motifs in ZIKV prM and E highlighted that *N*-glycosylation on the structural proteins was closely linked to the life cycle of ZIKV. In particular, the *N*-glycan on prM was important for the infectivity and cell-to-cell spread of the virus.

### 3.2. ZIKV prM and E N-Glycans Were Requried for Effective Secretion of the ZIKV E Protein

The results from the rescued ZIKV variants suggested that prM *N*-glycosylation might be involved in virus release. We hypothesized that prM *N*-glycosylation is important for virus protein trafficking and virus secretion; to investigate this we wanted to separate virus replication from the assembly events by using a virus-like particle system. 

The four *N*-glycosylation variants of prM and E genes were expressed from plasmid vectors in Vero B4 cells. To confirm the presence/absence of *N*-glycosylation, harvested lysates were treated with PNGase F and the prM and E proteins were analysed. As expected, the WT clone showed a downshift in molecular mass after PNGase F treatment, for both the prM and E proteins. N69Q showed a downshift only for the E protein, N154Q showed a downshift only for the prM protein, and N69Q/N154Q showed no downshift (Figure 4a).

To analyse the effect of *N*-glycosylation for the expression and secretion of E proteins, we harvested and analysed cell lysates and cell culture supernatants. In the cell lysate, there was no significant difference in expression levels of the E protein between the WT clone and N154Q, whereas N69Q and N69Q/N154Q showed about 43% and 56% reduction, respectively (Figure 4b,c). When analysing secretion into the supernatant, the WT clone had the highest level of secreted E protein, while N154Q showed 80% reduction of secretion even though the E protein expression in the cell lysate was similar to WT. The N69Q and N69Q/N154Q variants showed strong impairment of E protein secretion (86% and 95% reduction, respectively). The result highlighted that *N*-glycosylation of both prM and E was involved in E protein secretion.

### 3.3. Absence of the prM N-Glycan Caused Protein Aggregation

After we showed that the lack of prM *N*-glycan affected the secretion of the E protein, we hypothesized that the E protein might be trapped inside the ER due to the impaired secretion.

In order to verify the hypothesis, cells were transfected with the prME plasmid, and the ZIKV E protein localization inside cells was analysed together with the rough ER marker, Sec61α or an ERGIC marker, ERGIC53. The E-protein expressed from the WT prME construct showed limited co-localization with the rough ER but more co-localization with the ERGIC, which could be interpreted as efficient ER-Golgi trafficking (Figure 5). Interestingly, it seems as the E-protein from the N154Q mutant showed some localization with the rough ER compared with ERGIC, possibly indicating that E protein *N*-glycosylation was important for localization and trafficking in the cell. (Figure 5). On the other hand, the presence of *N*-glycosylation mutants in prM resulted in a high level of E protein co-localization with the rough ER. Compared to the WT and N154 mutant, the E protein seemed to be highly aggregated (Figure 5a). However, these aggregates did not co-localize with the ERGIC53 marker (Figure 5b) indicating that the glycosylation of prM was important for the intracellular trafficking out of ER. 

This data suggested that the *N*-glycan in prM was important for E protein trafficking and that the *N*-glycan in the E protein might partially be involved in the transport of assembled particles to Golgi by interacting with ERGIC53, which is a mannose-specific membrane lectin operating as a cargo receptor for the transport of glycoproteins from the ER to the ERGIC.

### 3.4. Lack of the prM N-Glycan Induced Nuclear Translocation of CHOP

No prM *N*-glycosylation led to apparent aggregation of the ZIKV E protein in the rough ER (Figure 5a). This indicated a disequilibrium of the ER protein homeostasis, probably due to accumulation of mis- or unfolded proteins leading to ER stress. One marker for ER stress is the upregulation of CHOP [34]. Thus, we decided to study if CHOP expression was affected by the absence/presence of the *N*-glycan motif in the prME protein. 

To induce ER stress conditions, we treated Vero B4 cells with Brefeldin A (1 µg/mL) (an inhibitor for protein transport from the Golgi apparatus to ER) and Tunicamycin (1 µg/mL) (an inhibitor for N*-*glycosylation synthesis), respectively. After 24 h, the cells were stained with CHOP antibody and examined from confocal microscopy. As expected, both inhibitor-treated cells showed a high level of CHOP in the nucleus (Figure 6). Next, we transfected the four different prME plasmids into Vero B4 cells and analyzed expression of the E protein and CHOP. The expression of WT prME did not result in CHOP expression. In contrast, transfection and the following expression of both *N*-glycosylation mutants (prM or E), resulted in CHOP expression and its nuclear translocation (Figure 6). 

Overall, the lack of *N*-glycosylation in the prM or E protein seemed to induce ER stress which suggested that *N*-glycans in the nascent prME protein was important for protein folding and/or transit through the ER.

## 4. Discussion

Asparagine-linked glycosylation (*N*-glycosylation) is a process when an oligosaccharide is attached to a newly synthesized protein [35] which is important to retain proper folding in nascent proteins [36]. ZIKV translates a single polyprotein from the genome and cleaves the polyprotein to several viral proteins [37]. Among viral proteins, only three proteins (prM, E, and NS1) interact with the enzymes involved in the *N*-glycosylation during the post-translational process, hence they possess an *N*-glycan motif (N-X-S/T) [38].

Several studies have investigated the role of *N*-glycosylation in the E protein for the ZIKV life cycle, in viral assembly and infectivity [23,24], tissue tropism [25], and pathogenesis [6,17]. Regarding the NS1 protein, *N*-glycosylation in NS1 was shown to be essential for DENV, WNV, and ZIKV to trigger endothelial hyper-permeability via clathrin-mediated endocytosis [39]. Recently, Yuan et al. showed that a single mutation in the ZIKV prM protein (S139N) contributes to fetal microcephaly and another report discussed the role of prM protein in viral pathogenicity and potential for use in ZIKV vaccine development [40,41]. Nevertheless, the role of prM *N*-glycosylation in the ZIKV life cycle is not clear. 

In our study, *N*-glycosylation in ZIKV prM was necessary for completion of the ZIKV life cycle. When the *N*-glycosylation site in prM was removed, the spread of virus to neighboring cells was abolished, and basically no infectious virus and a very low amount of virus-like particle secretion was detected. Trafficking inside the cell was also impaired. The results highlighted the role of prM *N*-glycosylation as essential in the ZIKV infection cycle and suggested that it had effect on virus release, as previously observed for prM *N*-glycosylation in DENV, and TBEV [26,28,29], and on trafficking, as discussed for *N*-glycosylation of JEV prM, that has been suggested to play a role in modulating the efficiency of virus release and trafficking between ER and Golgi [26]. Lack of *N*-glycosylation in the E protein was not as detrimental to ZIKV as for prM. However, removal of the *N*-glycosylation site on ZIKV E resulted in smaller plaques, decrease of viral shedding, and thus fewer progeny viruses, similar to previous studies [6,11].

Our results suggested that *N*-glycosylation of ZIKV prM is necessary for correct folding of the prM protein and possibly the E protein, and that this will promote transport of immature virions from the ER to the plasma membrane, similar to studies on JEV [26]. Interestingly, co-localization of the ZIKV E protein and ERGIC 53 was dependent on *N*-glycosylation. ERGIC 53 is an integral membrane protein localized in the ERGIC [42] and operates as a cargo receptor for the transport of glycoproteins from the ER to the ERGIC [43]. *N*-glycans have high mannose content, which could interact with the mannose-specific binding lectin ERGIC 53 [44]. In line with our results, it has been suggested that hetero-dimerization of prM and E in TBEV is essential for the E protein to reach its final native confirmation [45].

CHOP is induced in response to certain stressors and is a trigger for ER stress-related apoptosis [46,47,48]. We found a close link between aggregation of the ZIKV E protein in ER and absence of the prM *N*-glycan. Interestingly, CHOP was then induced, most probably due to stress induced protein aggregation in ER.

When secretory and membrane polyproteins are inserted into the rough ER, Sec61α plays a crucial role, as it is tightly associated with membrane-bound ribosomes and oligosaccharyl transferase (OST), [49]. We observed that lack of *N*-glycosylation in the prM protein caused a higher level of E protein aggregation, which mostly co-localized in sec61α-stained ER. This suggested that the *N*-glycan in the prM protein could be involved in correct folding of the nascent polyprotein but also in transporting of the synthesized protein. This transport involves a complex formation of the ribosome, the Sec61 protein-conducting channel and the OST [50]. Further studies regarding interaction between Sec61α, N-glycosylated prM protein, and OST is required. Recently, the OST complex was suggested as an important flavivirus host factor with potential use as an anti-flaviviral target. The OST complex catalyzes the N-linked glycosylation of nascent proteins and when the function of the OST complex was compromised, it inhibited flavivirus RNA replication [51]. Thus, it is important for flaviviruses to be properly N-glycosylated during infection of their host cell. Further studies are required to identify how the prM *N*-glycan is involved in, e.g., protein cleavage of ZIKV polyprotein or protein folding.

The *N*-glycan motifs in flaviviruses have interesting features. Many flavivirus strains vary regarding their E protein *N*-glycosylation motif. For instance, the historic African-lineage ZIKV strain (MR766) lacks the *N*-glycosylation motif at N154Q. In contrast, a contemporary Asian-lineage ZIKV strain (FP/10087PF/2013) retains this motif. Furthermore, some strains of WNV, St. Louis encephalitis virus, and Yellow fever virus also lack the motif (Table 2). However, for prM and NS1, the *N*-glycosylation motif is retained for all flaviviruses (Table 2), which supports our finding that introducing a mutation on the prM *N*-glycosylation site is detrimental and will inhibit the flavivirus life cycle.

In conclusion, our study demonstrated the role of the prM *N*-glycan in the ZIKV life cycle by showing that lack of prM *N*-glycosylation caused impairment in E protein expression and secretion, E protein aggregation in ER, and eventually CHOP up-regulation and nucleus translocation (Figure 7). Further investigation of the interaction between prM *N*-glycan and ER-localized protein complexes will facilitate the development of more powerful antiviral strategies for ZIKV and other flaviviruses.

## Figures and Tables

**Figure 1 viruses-12-00925-f001:**
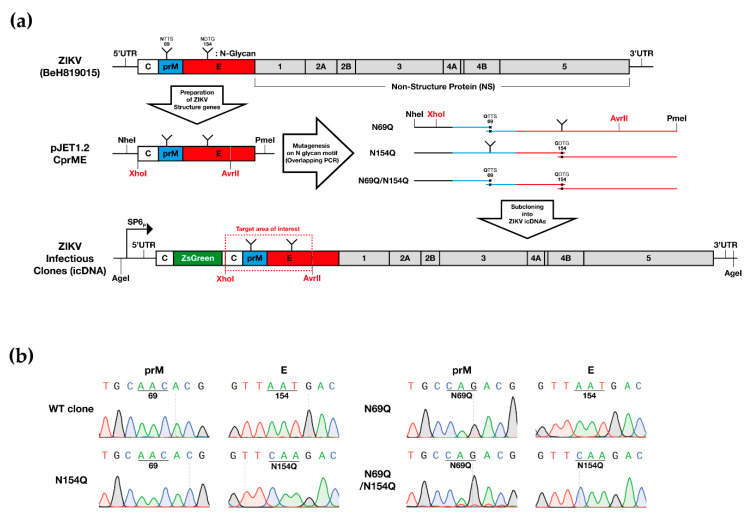
Generating *N*-glycosylation mutants of ZIKV BeH819015 in prM and/or E. (**a**) The structural genes of ZIKV BeH819015 were cloned into a pJET1.2 vector and *N*-glycan motif mutagenesis was introduced by overlapping polymerase chain reaction (PCR). Mutated PCR amplicons were sub-cloned into the infectious clone of ZIKV BeH819015 that contained a ZsGreen marker by using the indicated restriction endonuclease sites (red color). SP6 promoter sequence is present upstream of sequence corresponding to ZIKV genome; (**b**) Sequence chromatograms of the prM and E protein glycosylation motifs of ZIKV wild-type (WT), N69Q, N154Q or N69Q/N154Q clones. ZIKV = Zika virus; UTR = untranslated region; C = capsid protein; prM = pre-membrane protein; E = envelope protein; ZsGreen = green fluorescent protein derived from *zoanthus species*; WT clone = wild-type ZIKV with the ZsGreen reporter gene; N69Q = ZIKV with a mutated *N*-glycosylation site in prM; N154Q = ZIKV with a mutated *N*-glycosylation site in E; N69Q/N154Q = ZIKV with mutated *N*-glycosylation sites in prM and E.

**Figure 2 viruses-12-00925-f002:**
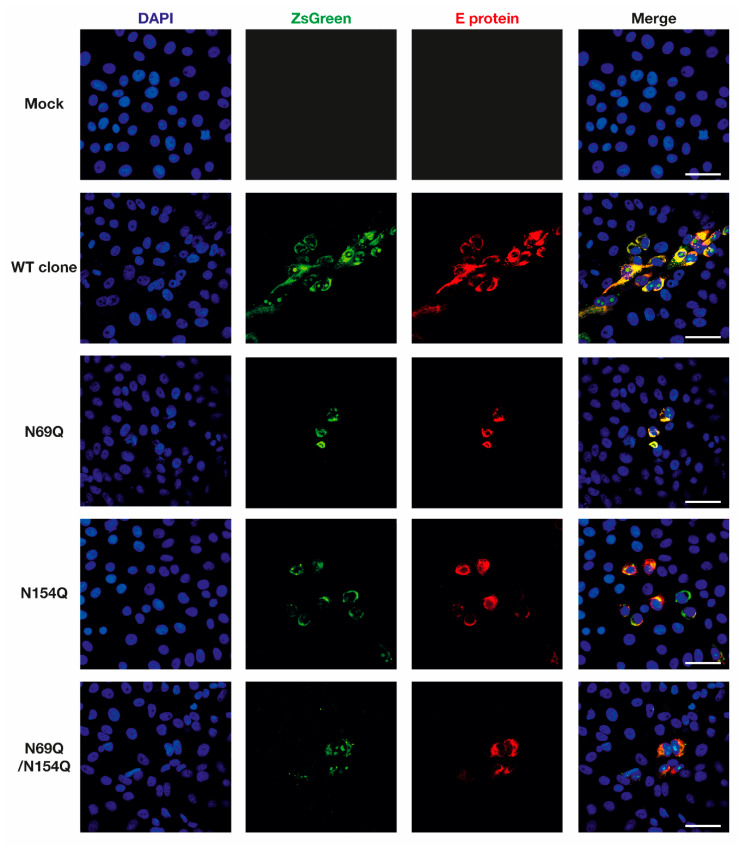
Confirmation of the inserted-marker (ZsGreen) and ZIKV E protein expression from rescued ZIKV variants. Vero B4 cells were grown on eight-well chamber slides. At 50% confluence, the cells were transfected with in vitro transcribed ZIKV RNA. At 72 hpt, the cells were fixed and stained with rabbit anti-ZIKV E antibody, anti-rabbit conjugated to Alexa Fluor 568 and DAPI. The samples were analyzed using a Nikon A1R+ confocal microscope and ImageJ software. Scale bar = 50 µm. DAPI = 4′,6-diamidino-2-phenylindole; ZsGreen = green fluorescent protein derived from *Zoanthus sp.*; E protein = envelope protein; WT clone = wild-type ZIKV with the ZsGreen reporter gene; N69Q = ZIKV with a mutated *N*-glycosylation site in prM; N154Q = ZIKV with a mutated *N*-glycosylation site in E; N69Q/N154Q = ZIKV with mutated *N*-glycosylation sites in prM and E.

**Figure 3 viruses-12-00925-f003:**
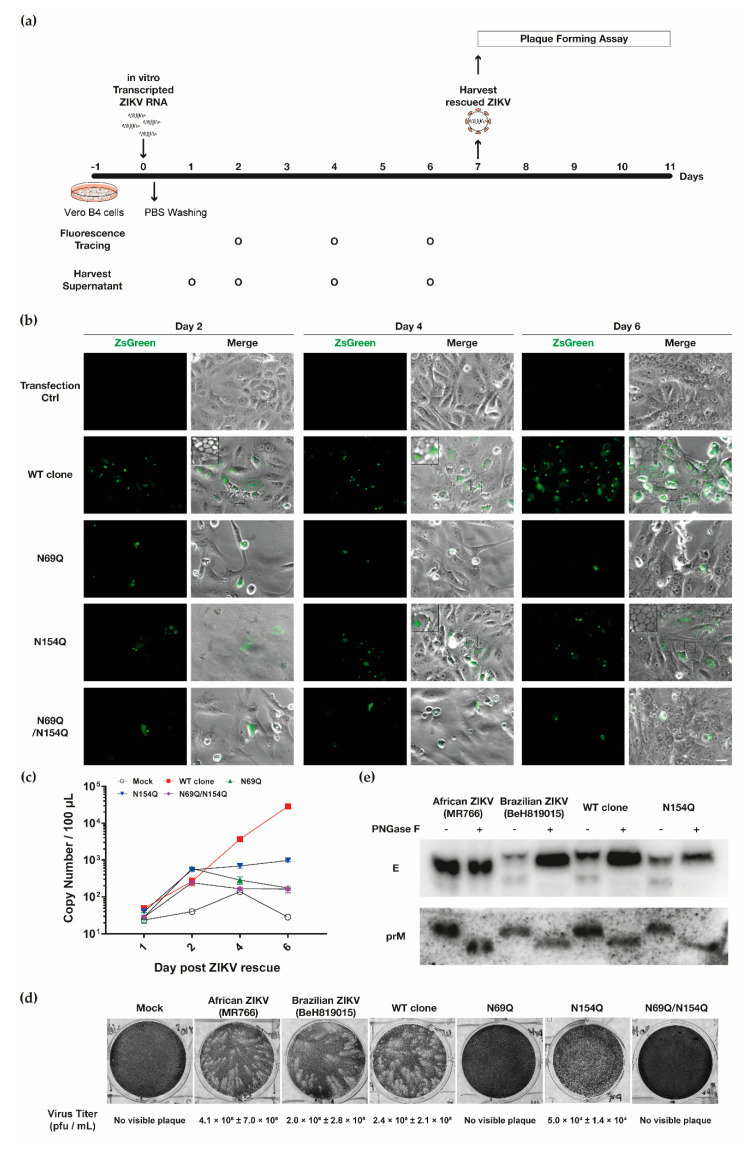
The *N*-glycan on ZIKV prM is important for the virus life cycle. (**a**) Schematic diagram of experiments. (**b**) Representative picture of ZsGreen expression from four variants of ZIKV RNA transfected into Vero B4 cells. The monitoring was performed as scheduled in Figure 3**a**. Scale bar = 20 µm. (**c**) Virus egress kinetics by qRT-PCR, depicted as copy number of ZIKV NS5 RNA in 100 µL of harvested supernatant, as scheduled in Figure 3**a**. Mean values and standard deviations are shown from three independent experiments. (**d**) Representative picture of the plaque-forming ability of harvested viruses. The ZIKV strains MR766 and BeH819015 (approximately, 100 plaque-forming units) were used as positive controls. Respectively, virus titer was measured using plaque assay and the average of virus titer and standard deviations are shown from two independent experiments. (**e**) Confirmation of presence/absence of the *N*-glycan sites in the prM and E proteins. Vero B4 cells were infected with supernatant from day 6 (Figure 3**a**). Three days after infection, the cell lysate was harvested and used for prM and E protein detection by Western blotting, with either absence or presence of peptide N-glycosidase F (PNGase F) treatment. The ZIKV strains MR766 (absence of *N*-glycan in E) and BeH819015 (presence of *N*-glycan in E) were used as controls.

**Figure 4 viruses-12-00925-f004:**
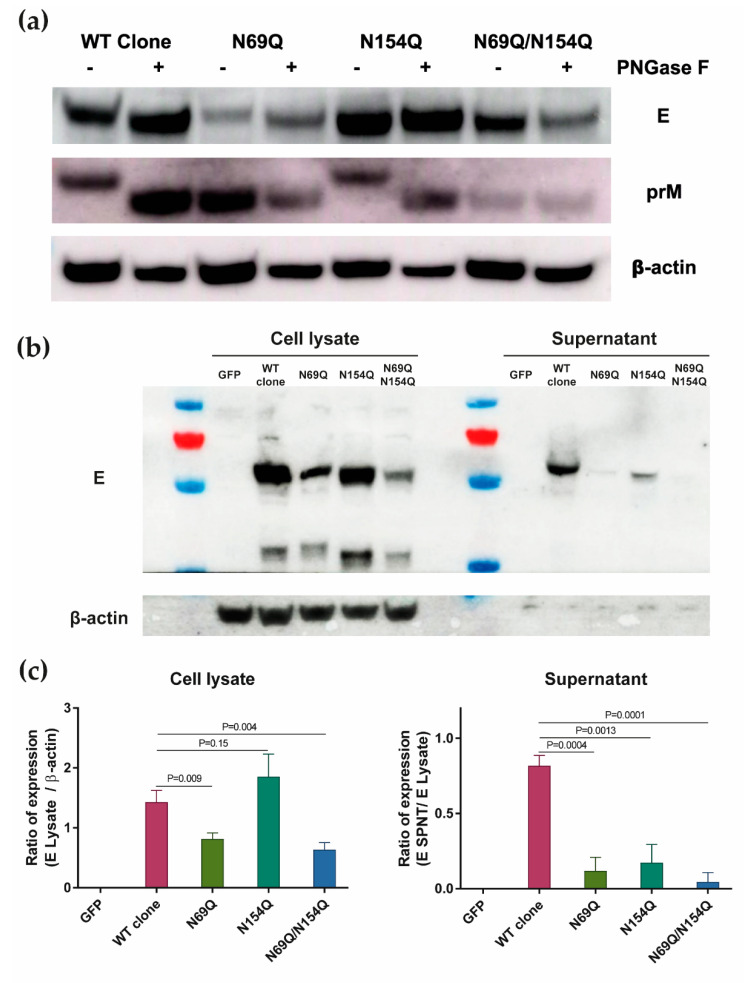
The lack of *N*-glycan on the pre-membrane (prM) or/and E protein caused impairment in the E protein expression and secretion. (**a**) Confirmation of absence/presence of *N*-glycosylation in the prM- and E proteins expressed from constructed plasmids. For the Western blotting, proteins were stained with anti-ZIKV E -, anti-ZIKV prM- or anti-β actin protein antibody (**b**) Representative Western blot picture of E- and β-actin protein expression in harvested cell lysate and cell culture supernatant from three independent experiments. (**c**) Quantification of E protein expression with Image J. To normalize the E protein expression in the cell lysate, the expression level of E protein was divided by the expression level of β-actin. To normalize the E protein expression in the supernatant, the expression level of E protein from the supernatant was divided by the expression level of E protein from the cell lysate. The experiment was repeated three times, average and SD are shown. Statistical significance was determined by two-tailed t-test. *p* = *p* value. GFP = pcDNA3.1-GFP vector; WT clone = pcDNA3.1-prME vectors with wild-type ZIKV prM-E genes; N69Q = pcDNA3.1-prME vector with a mutated *N*-glycosylation site in prM; N154Q = pcDNA3.1-prME vector with a mutated *N*-glycosylation site in E; N69Q/N154Q = pcDNA3.1-prME vector with mutated *N*-glycosylation sites in prM and E; PNGase F = Peptide:N-glycosidase F.

**Figure 5 viruses-12-00925-f005:**
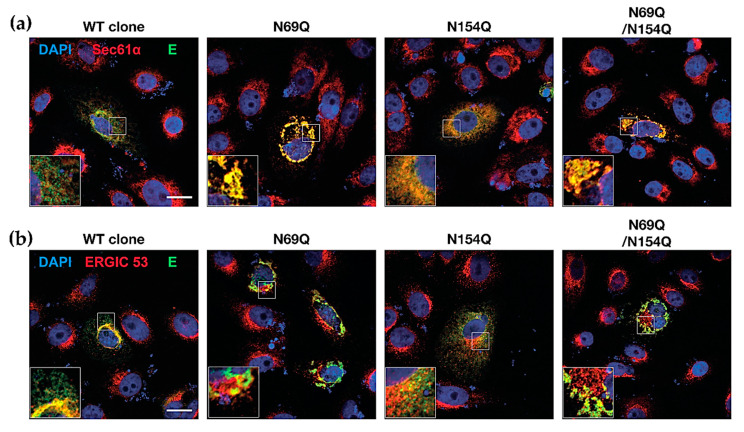
Co-localization of the E protein expression with various cellular protein markers. Vero cells were transfected with pcDNA3.1-prME plasmids. (**a**) The immunofluorescence microscopy images of ZIKV E-Sec61α double-stained Vero B4 cells. After 48 h, cells were stained with a rabbit anti-ZIKV E protein antibody and a mouse anti-Sec61 alpha antibody followed by secondary antibody and DAPI. (**b**) The immunofluorescence microscopy images of ZIKV E-ERGIC (ER-Golgi intermediate compartment) 53 double-stained Vero B4 cells. After 48 h, cells were stained with a rabbit anti-ZIKV E protein antibody and a mouse anti-ERGIC 53 antibody, followed by secondary antibody and DAPI. The samples were analyzed using a Zeiss 710 confocal microscope and ImageJ software. Scale bar = 20 µm. Sec61α = Protein transport protein Sec61 subunit alpha; ERGIC 53 = ER-Golgi intermediate compartment 53 kDa protein or lectin mannose-binding 1.

**Figure 6 viruses-12-00925-f006:**
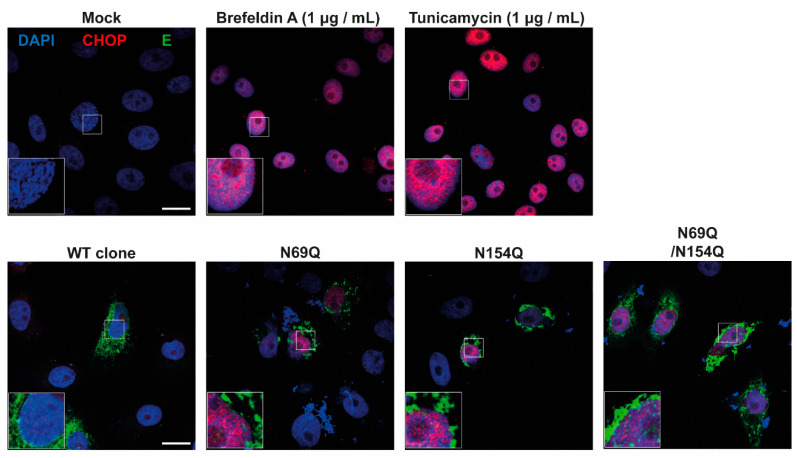
The lack of *N*-glycan on the prME protein induced ER stress. For the immunofluorescence assay, Vero cells were transfected with pcDNA3.1-prME plasmids for 48 h, samples were stained with a mouse anti-CHOP (CCAAT-enhancer-binding protein homologous protein) antibody together with a rabbit anti-ZIKV E antibody followed by secondary antibodies and DAPI. In positive controls, CHOP expression was induced with Brefeldin A (1 µg/mL) or Tunicamycin (1 µg/mL) for 24 h. Representative images of ZIKV E-CHOP double-stained Vero B4 cells. Scale bar = 20 µm.

**Figure 7 viruses-12-00925-f007:**
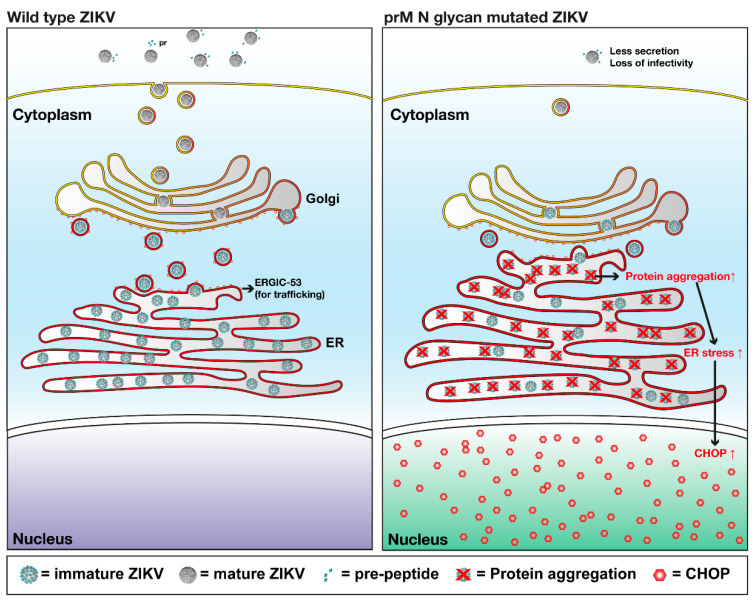
Hypothetical model of the role of *N*-glycosylation of the prM protein in ZIKV. Mutation in the prM *N*-glycosylation site caused accumulation of virus proteins in ER, which induced increased ER stress in the cell. Eventually, the up-regulated ER stress triggered the expression of CHOP and its translocation into nucleus. ZIKV = Zika virus; Golgi = golgi apparatus; ER = endoplasmic reticulum; ERGIC-53 = ER-Golgi intermediate compartment 53 kDa protein or lectin mannose-binding 1; prM = pre-membrane protein; CHOP = C/EBP Homologous Protein.

**Table 1 viruses-12-00925-t001:** List of primers used to mutate the *N*-glycan motif in the ZIKV BeH819015 pre-membrane and/or envelope protein.

Name	Sequence (5′→3′)
F-NheI_C	CA **GCTAGC**^1^ATGAAAAACCCAAAAAAGAAATCC
R-PmeI_E	TA **GTTTAAAC**^1^TTAAGCAGAGACGGCTGTGGA
F-N69Q	GATTGTTGGTGCCAGACGACGTCA
R-N69Q	TGACGTCGTCTGGCACCAACAATC
F-N154Q	GGGATGATCGTTCAAGACACAGGA
R-N154Q	TCCTGTGTCTTGAACGATCATCCC

^1^ The introduced restriction enzyme sites are highlighted in bold text.

**Table 2 viruses-12-00925-t002:** The comparison of *N*-glycan motifs in different flaviviruses.

Viruses (Strain)	*N*-glycan Motifs		
	prM	Env	NS1
WNV (NY99)	N15	N154	N130, N175, N207
WNV (ArB3573/82)	N15	-	N130, N175, N207
JEV (SA-14)	N15	N154	N130, N207
SLEV (MS1-7)	N15	-	N130, N175, N207
DENV1 (Nauru/West Pac/1974)	N69	N67, N153	N130, N207
DENV2 (Thailand/16681/1984)	N69	N67, N153	N130, N207
DENV3 (Sri Lanka/1266/2000)	N69	N67, N153	N130, N207
DENV4 (Singapore/8976/1995)	N69	N67, N153, N472	N130, N207
ZIKV (MR766)	N69	-	N130, 207
ZIKV (FP/10087PF/2013)	N69	N154	N130, 207
YFV (17D vaccine)	N13, N29	-	N130, 208
YFV (Ivory coast/1999)	N13, N29	-	N130, 208
TBEV (Hypr)	N27	N154	N85, N207, N223
POWV (LB)	N27	N154	N85, N207, N223

prM = pre-membrane protein; Env = envelope protein; NS1 = non-structural protein 1; WNV = West Nile fever virus; JEV = Japanese encephalitis virus; SLEV = St. Louis encephalitis virus; DENV = Dengue virus; ZIKV = Zika virus; YFV = Yellow fever virus; TBEV = Tick-borne encephalitis virus; POWV = Powassan virus.

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
