# Peer review of "N-glycosylation in the Pre-Membrane Protein Is Essential for the Zika Virus Life Cycle"

_viruses, 2020, doi:10.3390/v12090925_

Round 1

Reviewer 1 Report

The manuscript titled “N-glycosylation in pre-membrane protein is essential for the Zika virus life cycle” details a series of efforts examining the importance of membrane protein glycosylation to Zika virus infectivity. Using a series of microscopy and western blotting techniques the role of the N69 and N154 glycosylation sites are assessed. The mutation of N69 results in severe attenuation and the activation of an ER-stress response in transfected cells. Further, the N154 mutation is impaired in several regards, consistent with the literature in the field. On the whole the study is thorough, and is very tempered in their conclusions and interpretation of the data.

There are several noted concerns, but none of these are essential to the interpretation of the study as submitted.

Concerns-

  • Did the study further examine the processing of prM into M and Pr? Was this processing negatively impacted by the modification of the N69 site?

  • What is the extent to which CHOP / ER stress activation is responsible for limiting the production of infectious virions?

  • Was there differences in the amount of viral capsid protein released in the single and double mutant particles?

Author Response

Reviewer 1

QUESTION: Did the study further examine the processing of prM into M and Pr? Was this processing negatively impacted by the modification of the N69 site?

ANSWER: This is a very interesting question, but we did not have time/resources to examine this. It will be important in future studies. The processing of prM into M and pr depends on furin activity at the specific cleavage site (R91, R-X-X-R), which is close to the N glycan motif in prM (N69, N-X-S/T). We don’t know whether the prM N glycan mutation affects furin activity.

QUESTION: What is the extent to which CHOP / ER stress activation is responsible for limiting the production of infectious virions?

ANSWER: The C/EBP homologous protein (CHOP, also known as GADD153) is a transcription factor that is activated at multiple levels during ER stress. The CHOP activation could induce cell death by promoting protein synthesis and oxidation (Stefan J. Marciniak et al,. Genes and Development. 2014).
In our study, we found that the prM N-glycan mutation caused ER aggregation of the E protein and nuclear translocation of the CHOP protein. Based on these observed results, we hypothesized that if the cells are forced to keep prM N-glycan mutated proteins inside due to lack of ability for trafficking and secretion, then, eventually, the cells will die. This hypothesis need further investigation, but is beyond the scope of this study.

QUESTION: Was there differences in the amount of viral capsid protein released in the single and double mutant particles?

ANSWER: This is also a very interesting question. At the beginning of this study, we constructed the “Capsid-pre-membrane-envelope” expression vector and transfected it to mammalian cells. Then, we found that it had a toxic effect, most probably because the capsid protein was not converted to the active form without cleavage by the viral “NS2B-NS3 protease”. When we discovered this toxic effect, we decided to not include viral capsid protein expression in our VLP system.

So far, we have not checked the amount of viral capsid protein after the introduction of the single and double mutant. This topic can be investigated in future studies, e.g. (i) Is there a role of the N-glycan mutation in prM or E for viral capsid protein expression and secretion? (ii) Can the N-glycan mutation in prM or E interfere with the activity of NS2B-NS3 viral protease?

Reviewer 2 Report

Gwon et al worked on the importance of the N-glycosylation in pre-membrane protein to the Zika virus. This is a well worked out theme among various viruses. The data agreed well with the current paradigm and seem incremental. To add more discoveries, it might be important to address some extra points to enrich the discoveries so that it makes a significant, nontrivial contribution. Some points are listed below.

  1. 1 / ln 43, is post-Golgi refers to trans-Golgi network?
  2. 4-5, RT-PCR for quantification seems to need an internal control for normalization and comparison among differentially treated cells. Say actin mRNA? Can ddPCR be used instead for better quantification?
  3. Pg 5 / ln 176, 105 should be 10^5.
  4. Pg 5 / ln 177, what is the titer number for rescued viruses?
  5. Pg 5 / ln 204-208, is the actin band used as an internal control for data analysis?
  6. 6 / ln 223, why one-way ANOVA? Two-way ANOVA should be used for high-stringency.
  7. 2, why was the infection rate so low, even for WT?
  8. 3 and pg 8 / ln 250, lack of ER-vacuolization is not well demonstrated. A marker is needed or EM examination is required to show. A positive control could help strengthen it as well.
  9. 3c, the number of viruses produced from the N69Q and N69Q/N154Q mutants are ~ 500 folds lower than the WT, but still significantly higher than mock. It is not reliable to conclude that these mutants are not producing any progeny.
  10. Ln 282-283, what is the dynamic range of the detection in the plaque counting? Can a 500-fold difference be reliably detected as in Fig. 3c?
  11. A control could be that the WT-viruses treated with PNGase F in Fig 3d so that the virus titer could be similar.
  12. 4 b/c, comparison of prM proteins would be needed.
  13. 5, and Pg 13 / ln348-350, regarding the N154 and ERGIC53, the colocalization is not strong or pretty diffuse and weak. It needs a better experiment or EM examination to better evaluate the point.
  14. The proposed aggregation of prM and the interaction between prM/E based on Fig. 5 needs better data. Are they aggregating in membranes? Or oligomerization? Or an adapter protein brings them together?
  15. Fig 6. CHOP is not strong in the ER any more after the treatment for 24 hrs. How about a time-course to show? Or RT-PCR to test the transcription time-course? Does a mutant NS1 do the same?
  16. 15, ln 403, NO PROGENY seems to go against Fig. 3c.
  17. Fig 7. the conclusion about mis-folded prM in ER is not supported by data. N69Q prM is stuck in the ER and expression is lower. Misfolding is one possibility. Alternatively, maybe an adapter protein needs to recognize its N-glycan to facilitate its egress from ER. Or maybe an ER retention signal is not properly shielded in the mutant even though it is folded properly. Or the mutant might be retrieved back much better from Golgi.
  18. The oligomerization of prM and the heterodimer of prM and E are proposed in different parts. The two points need high-resolution EM and /or biochemical data to demonstrate.

Author Response

Reviewer 2

QUESTION: 1 / ln 43, is post-Golgi refers to trans-Golgi network?

ANSWER: We appreciate your suggestion. In the revised manuscript, Pg.1 / In 43 is changed to trans-Golgi network.

QUESTION: 4-5, RT-PCR for quantification seems to need an internal control for normalization and comparison among differentially treated cells. Say actin mRNA? Can ddPCR be used instead for better quantification?

ANSWER: We agree with your suggestion that the RT-PCR ideally would have internal control for normalization. However, in this study, we used the RT-PCR to measure the copy number of ZIKV NS5 RNA in the supernatant. By measuring the viral genome in the supernatant, we could only measure secreted virus and since cellular material is basically absent in the supernatant, we believe that measuring an internal control for normalization is not suitable for isolated total RNA from the supernatant. Regarding ddPCR, it is a very sensitive technique, however, we would need a different technical set-up at our lab (other machines, etc). Thus, although ddPCR has higher sensitivity, it is not possible to perform in our lab.

QUESTION: Pg 5 / ln 176, 105 should be 10^5.

ANSWER: We appreciated your suggestion. In the revised manuscript -  Pg.5 /line 180 in the version with track-changes -  it is changed to 105.

QUESTION: Pg 5 / ln 177, what is the titer number for rescued viruses?

ANSWER: The virus titers were calculated and added to the manuscript.

Virus

Virus Titer (pfu/ml)

Mock

No visible plaques

African ZIKV (MR766)

4.10 x 106 ± 7.07 x 105

Brazilian ZIKV (BeH819015)

2.00 x 106 ± 2.83 x 105

WT clone

2.45 x 106 ± 2.12 x 105

N69Q

No visible plaques

N154Q

5.00 x 104 ± 1.41 x 104

N69Q/N154Q

No visible plaques

In the revised manuscript, virus titers have now been included as stated below.

Lines 150-152: “In parallel, the same above protocol was used for virus titration except viruses were 10-fold serial diluted (from 10-1 to 10-5). Visible plaques were counted and the viral titers (plaque-forming unit (pfu) / ml) were calculated.

Pg.9-10 / Figure 3 and figure 3 legend is changed.

Pg. 10 / at line 287 “and the virus titer was 49 fold lower (Figure 3d)” was added.

QUESTION: Pg 5 / ln 204-208, is the actin band used as an internal control for data analysis?

ANSWER: Yes, the actin band was used as an internal control for data analysis, and the result are shown in Figure 4bc. In the revised manuscript, pg.5 / ln 208 is changed.

QUESTION: 6 / ln 223, why one-way ANOVA? Two-way ANOVA should be used for high-stringency.

ANSWER: The reason for selecting one-way ANOVA for statistical analysis is because we only had one independent variable (see results in Figure 4c). We agree on your comment that two-way ANOVA could provide high-stringency, however, two-way ANOVA should be used when there are two independent variables.

QUESTION: 2, why was the infection rate so low, even for WT?

ANSWER: Thank you for the insightful comment. The reason for the relatively low infection rate in Figure 2 is because here it is transfection of in vitro transcripted viral RNA (rescue). In our hands, it usually took 6 days to reach a plateau of detectable infection and the result in Figure 2 was at 72 hours (3 days) post-transfection.

This is the infection/transfection rate of the WT clone at Day 3 compared to Day 6 (see attached pdf)

WT clone – Day 3

WT clone – Day 6

QUESTION: 3 and pg 8 / ln 250, lack of ER-vacuolization is not well demonstrated. A marker is needed or EM examination is required to show. A positive control could help strengthen it as well.

ANSWER: We agree with your comment that lack ER-vacuolization could not be conclusively demonstrated. We suspect lack of ER-vacuolization in Fig 3b (presence/absence of ER-vacuolization in WT clone compared to N154Q) could be an additional sign of ZIKV infection – see Monel et al,. EMBO Journal, 2017. However, since it is not well demonstrated we remove the red arrows in figure 3 and removed the sentence in the figure 3 legend describing this.

Also: Page 8-9 / lines 252-254 the statements on ER-vacuolization have been deleted. 

Also: Page 10 / lines 260-261 in fig 3 legend was changed to delete ER-vacuolization statements.

QUESTION: 3c, the number of viruses produced from the N69Q and N69Q/N154Q mutants are ~ 500 folds lower than the WT, but still significantly higher than mock. It is not reliable to conclude that these mutants are not producing any progeny.

ANSWER: Thank you for the comment and you are right. We could observe secretion of virus RNA from the N69Q and N69Q/N154Q mutants in figure 3c. The more important finding was that even though there was secretion of progeny virus RNA, the viruses were non-infectious (shown in figs 3c and d).

In the revised manuscript on page 15, line 409 is changed from “progeny virus” to “infectious virus”.

QUESTION: Ln 282-283, what is the dynamic range of the detection in the plaque counting? Can a 500-fold difference be reliably detected as in Fig. 3c?

ANSWER: The detection range for plaque counting was based on the dilution range from 10-1 to 10-5 of the original virus stock. In our hands, N69Q and N69Q/N154Q had no infectivity at all. The 500-fold difference in Fig 3c is based on detection of virus RNA from harvested supernatant, not infectious virus. To study if the virus RNA detected was infectious or non-infectious, we performed the plaque-forming assay presented in Fig 3d. However, our opinion is that we cannot directly compare the result from figure 3c with figure 3d.

QUESTION: A control could be that the WT-viruses treated with PNGase F in Fig 3d so that the virus titer could be similar.

ANSWER: This is an interesting suggestion. However, one concern is that PNGase F may also affect other N-glycans on e.g. the cell surface and thus we would not be certain of the result.

QUESTION: 4 b/c, comparison of prM proteins would be needed.

ANSWER: When we performed IF (fig 5 and 6), the prM antibody did not detect prM, thus we only used the E-antibody in figure 4 b and c. We believe that it is enough to detect the E protein since we wanted to study the secretion of VLPs and compare that to complete virus secretion. Thus, we used E as a marker for that purpose.

QUESTION: 5, and Pg 13 / ln348-350, regarding the N154 and ERGIC53, the colocalization is not strong or pretty diffuse and weak. It needs a better experiment or EM examination to better evaluate the point.

ANSWER: Thank you for the comment. We agree that the interpretation of the images can be difficult and an EM examination may help. However, although EM will show a better resolution inside the cell, we would still need to figure out how to pinpoint the exact area where the interaction takes place between prM-E and ERGIC53. We believe that this approach could be another project.

The co-localization in fig 5 is not strong, instead more diffuse when looking at N154 and ERGIC53. It is likely because of lack of the N-glycan in the E protein, which makes it less likely to interact with ERGIC53 (a mannose-specific membrane lectin and N glycan are consists of high mannose glycan). We have performed this experiment more than 3 times, with similar results. Thus, we believe that it is relevant to keep figure 5 as it is, but we have amended the text in the manuscript.

We have changed to: “Interestingly, it seems as the E-protein from the N154Q mutant showed some localization with the rough ER compared with ERGIC, possibly indicating that E protein N-glycosylation was important for localization and trafficking in the cell (Figure 5).”

QUESTION: The proposed aggregation of prM and the interaction between prM/E based on Fig. 5 needs better data. Are they aggregating in membranes? Or oligomerization? Or an adapter protein brings them together?

ANSWER: Thank you for the good question and suggestion. Currently, we suggest that the aggregation is in the ER-membrane, but we don’t know whether there is oligomerization or involvement of an adapter protein in this aggregation. This also could be a good direction for another ZIKV study. We have already discussed regarding potential adaptor proteins (OST complex and Sec61a) in the Discussion section (pg 15 / ln 434-438).

QUESTION: Fig 6. CHOP is not strong in the ER any more after the treatment for 24 hrs. How about a time-course to show? Or RT-PCR to test the transcription time-course? Does a mutant NS1 do the same?

ANSWER: In figure 6, we believe that CHOP is always at low levels in the ER or nucleus in mock experiments, because there is no reason for CHOP to be strong in normal cells, thus we did not perform a time-course experiment. We performed RT-PCR to detect CHOP transcription after 24 hours. The level of CHOP transcription was significantly up-regulated only in control groups (Brefeldin A and Tunicamycin) (data not shown). The question regarding the possible effect of a mutant NS1 is very interesting, but beyond the scope of our study. However, we found one paper that might be able to explain the question (Milly M. Choy et al,. A Non-structural 1 Protein G53D Substitution Attenuates a Clinically Tested Live Dengue Vaccine, Cell Reports, 2020)

QUESTION: 15, ln 403, NO PROGENY seems to go against Fig. 3c.

ANSWER: We agree with this comment and we have removed “progeny virus” and replaced it with “infectious virus” instead (line 409).

QUESTION: Fig 7. the conclusion about mis-folded prM in ER is not supported by data. N69Q prM is stuck in the ER and expression is lower. Misfolding is one possibility. Alternatively, maybe an adapter protein needs to recognize its N-glycan to facilitate its egress from ER. Or maybe an ER retention signal is not properly shielded in the mutant even though it is folded properly. Or the mutant might be retrieved back much better from Golgi.

ANSWER: We agree with the reviewer’s comment and have removed “mis-folded” from figure 7 and changed it to “protein aggregation”.  In the Fig 7 legend we have removed “mis-folded”.

QUESTION: The oligomerization of prM and the heterodimer of prM and E are proposed in different parts. The two points need high-resolution EM and /or biochemical data to demonstrate.

ANSWER: We agree with the reviewer’s comment. We hypothesized that there will be misfolded prM in ER based on literature. Because we learned that the N-glycan is usually involved in protein folding and quality control of nascent protein. However, we never showed that this was the case in our study. Thus, in figure 7, the explanation about mis-folded prM in ER was removed. In the revised manuscript, the title of figure 7 is changed to: “Hypothetical model of the role of N-glycosylation of the prM protein in ZIKV”, from “Diagram depicting the role of N-glycosylation of the prM protein in ZIKV”.

Reviewer 3 Report

In this manuscript, the authors describe their study of the role of ZIKV prM and E protein N-linked glycosylation in the viral life cycle. They use an in-house infectious clone from a contemporary strain of ZIKV to generate mutant viruses with the single N-linked glycan sites in prM and E protein disrupted individually or together. As previously published, the E protein modification is dispersible for spread in Vero cells, but the prM mutant was highly attenuated. They further define how this mutation impairs viral particle release, likely by aggregation of prM in the ER-Golgi, which results in ER stress.

This is not a new topic, especially since E protein glycosylation has been well studied. The impact of loss of glycosylation in prM is still less studied, and thus novel in this manuscript. The manuscript is well written, and the experiments are mostly convincing.

One major issue is the lack of quantification of mutational impacts on viral titers. Most infection experiments are merely qualitative. Absolute titers of rescued virus should be reported to show the degree of impairment by these mutations.

A minor comment is in regards to Fig 3e. The E protein bands blot needs to be run further to clearly show size differences. Also, a loading control could be included.

Author Response

Reviewer 3

QUESTION: One major issue is the lack of quantification of mutational impacts on viral titers. Most infection experiments are merely qualitative. Absolute titers of rescued virus should be reported to show the degree of impairment by these mutations.

ANSWER: We appreciate your concern and agree with your comment. In the revised manuscript, the titers of rescued virus are included in Figure 3d

QUESTION: A minor comment is in regards to Fig 3e. The E protein bands blot needs to be run further to clearly show size differences. Also, a loading control could be included.

ANSWER: Thank you for the comment. We have tried to run the gel further several times, also with different composition of acrylamide and running buffer. However, it was not easy to get clarification of the size differences since the size of N-glycan corresponds to approximately 2-3 KDa, which represents 55 of the E protein (56 KDa). For prM (23KDa), the N-glycan is ca 10% of the weight and thus easier to differentiate on the gel, although we believe that we can still detect the difference for the E-protein as well.

The purpose of the western blot in Figure 3e was to show the absence/presence of N-glycan from rescued viruses and there was no quantification involved. Thus, we did not consider to include internal control in that experiment.

Round 2

Reviewer 2 Report

The authors have addressed most of my concerns. There are a few points to be refined. I listed them here. 

1) For the one-way ANOVA, the basic assumption is the same standard deviation for the different groups of measurements. Data in Fig. 4C appear not to satisfy this requirement. It might be better to use standard t-test.

2) Ln 237-239, the correlation between ZsGreen and E expression in the infected cells seems to be stretched to show the extent of genome replication. The authors want to believe that there is a linear (or at least monotonic) correlation between the two. But no data were showed to demonstrate that the higher ZsGreen expression is, the more replicated copies of viral genome.

3). Fig. 3C shows that the N60Q mutant, when compared to the wild-type, has ~0.2% progeny viruses released to supernatant), and these progenies were not infectious (Fig. 3D). Later it was showed that the virion particles were not able to move as well as wild-type to the Golgi complex and trans Golgi network. However, the exocytosis seemed to be intact as long as the virion particles reach the trans-Golgi network. The red cross in the right panel of Fig. 7 seems to be inconsistent with these observations. The production of mature, properly assembled virus particles is much lower, not the exocytotic release being blocked.

4) ln 409-410. The virus-like particle secreted to the medium is reduced to ~0.2% of wild-type as in figure 3C, not completely undetectable. It is also too exact to say the "trafficking inside the cell was abrogated", which means completely blocked. It needs some down-tuning. 

5) ln 416, "less progeny virus" should be "fewer progeny viruses".  "egress" of viruses was used in multiple places, which should be clarified to mean "exocytosis of the virions from the trans-Golgi network", if I understood correctly. Alternatively, it could mean the viral progeny being produced from the cells and released into the medium without pointing to the step of exocytosis.

Author Response

Question 1: For the one-way ANOVA, the basic assumption is the same standard deviation for the different groups of measurements. Data in Fig. 4C appear not to satisfy this requirement. It might be better to use standard t-test.

Answer: We appreciate your comment on the statistical analysis of data in Fig. 4C. Your point is right that it is better to use the standard t-test. We just have to mention that the standard t-test is usually considered when you have only two groups to check the statistical difference between groups. The data in Fig. 4C consist of 5 groups, with one variable (“Ratio of expression (E Lysate / β-actin)”) applied to all groups. That’s our reason to use one-way ANOVA with Dunnett’s post-hoc analysis for multiple comparisons. However, since we agree with your point, we re-analysed the data in Fig. 4C using standard t-test as you suggested.

The manuscript is revised with below changes,

Pg6 / ln 226-228 / Material and Methods – 2.10. Data analysis / The description was changed to “All statistical analyses were performed using two-tailed t-test by GraphPad Prism 7.0 software”.

Pg12 / ln 333, Figure 4 legend was changed to “statistical significance was determined by two-tailed t-test. P= p value.”

Pg12 / Figure 4C was modified accordingly and the actual p value is now presented in the graph.

Question 2: Ln 237-239, the correlation between ZsGreen and E expression in the infected cells seems to be stretched to show the extent of genome replication. The authors want to believe that there is a linear (or at least monotonic) correlation between the two. But no data were showed to demonstrate that the higher ZsGreen expression is, the more replicated copies of viral genome.

Answer: This is an interesting point and we appreciate your advice on the manuscript. As you pointed out, we did not show the direct comparison of ZsGreen expression with viral genome replication. However, the ZsGreen expression is correlated to the ZIKV infection site, as we also showed in a previous publication that used the same ZIKV WT clone in an antiviral assay (PMID=32235763).

Thus, we revised the manuscript.

Pg 6 / ln 239 / the statement was changed by removing “genome replication” and instead adding “protein synthesis and the site of viral infection”.

Question 3:  Fig. 3C shows that the N60Q mutant, when compared to the wild-type, has ~0.2% progeny viruses released to supernatant), and these progenies were not infectious (Fig. 3D). Later it was showed that the virion particles were not able to move as well as wild-type to the Golgi complex and trans Golgi network. However, the exocytosis seemed to be intact as long as the virion particles reach the trans-Golgi network. The red cross in the right panel of Fig. 7 seems to be inconsistent with these observations. The production of mature, properly assembled virus particles is much lower, not the exocytotic release being blocked.

Answer: We agree with your comment. The red cross in the right panel in Fig. 7 did not correctly represent the observed results. Thus, we modified figure 7 by removing the red cross and a new figure 7 was added to the manuscript at page 17.

Question 4: ln 409-410. The virus-like particle secreted to the medium is reduced to ~0.2% of wild-type as in figure 3C, not completely undetectable. It is also too exact to say the "trafficking inside the cell was abrogated", which means completely blocked. It needs some down-tuning.

Answer: You are right. The previous statement on the effects of prM N glycosylation mutation was exaggerated. As you suggested, we did some down-tuning in the revised manuscript.

Page 15 / ln 411-412, the statement with no virus-like particle and trafficking inside the cell was changed by removing “abrogated” and adding “a very low amount of virus-like particle secretion was detected. Also trafficking inside the cell was impaired”.

Question 5: ln 416, "less progeny virus" should be "fewer progeny viruses".  "egress" of viruses was used in multiple places, which should be clarified to mean "exocytosis of the virions from the trans-Golgi network", if I understood correctly. Alternatively, it could mean the viral progeny being produced from the cells and released into the medium without pointing to the step of exocytosis.

Answer: We appreciate your suggestion and we agree with the meaning pointing to the viral progeny being produced from the cells and released into the medium. As suggested, we amended the manuscript.

Page 10 / ln 274-275, “We then analyzed the kinetics of viral egress, by measuring the virus RNA copy numbers in the harvested supernatant…” was modified to “We then analyzed the kinetics of viral egress, here defined as viral progeny being produced from the cells, by measuring the virus RNA copy numbers in the harvested supernatant”

Page 15 / ln 418-419, the sentence “However, removal of the N-glycosylation site on ZIKV E resulted in smaller plaques, decrease of viral egress, and thus less progeny virus, similar to previous studies” was modified to:  “However, removal of the N-glycosylation site on ZIKV E resulted in smaller plaques, decrease of viral shedding, and thus fewer progeny viruses, similar to previous studies”

Page 15 / ln 414, “egress” was changed to “release”

Reviewer 3 Report

The authors adequately addressed my prior comments.

Author Response

Thank you for your statement